# Biomechanical and Clinical Effect of Patient-Specific or Customized Knee Implants: A Review

**DOI:** 10.3390/jcm9051559

**Published:** 2020-05-21

**Authors:** Jin-Ah Lee, Yong-Gon Koh, Kyoung-Tak Kang

**Affiliations:** 1Department of Mechanical Engineering, Yonsei University, 50 Yonsei-ro, Seoul 03722, Korea; gna0812@gmail.com; 2Department of Orthopaedic Surgery, Yonsei Sarang Hospital, 10 Hyoryeong-ro, Seocho-gu, Seoul 06698, Korea; osygkoh@gmail.com

**Keywords:** patient-specific, knee joint, customized implant, total knee replacement

## Abstract

(1) Background: Although knee arthroplasty or knee replacement is already an effective clinical treatment, it continues to undergo clinical and biomechanical improvements. For an increasing number of conditions, prosthesis based on an individual patient’s anatomy is a promising treatment. The aims of this review were to evaluate the clinical and biomechanical efficacy of patient-specific knee prosthesis, explore its future direction, and summarize any published comparative studies. (2) Methods: We searched the PubMed, MEDLINE, Embase, and Scopus databases for articles published prior to 1 February 2020, with the keywords “customized knee prosthesis” and “patient-specific knee prosthesis”. We excluded patient-specific instrument techniques. (3) Results: Fifty-seven articles met the inclusion criteria. In general, clinical improvement was greater with a patient-specific knee prosthesis than with a conventional knee prosthesis. In addition, patient-specific prosthesis showed improved biomechanical effect than conventional prosthesis. However, in one study, patient-specific unicompartmental knee arthroplasty showed a relatively high rate of aseptic loosening, particularly femoral component loosening, in the short- to medium-term follow-up. (4) Conclusions: A patient-specific prosthesis provides a more accurate resection and fit of components, yields significant postoperative improvements, and exhibits a high level of patient satisfaction over the short to medium term compared with a conventional prosthesis. However, the tibial insert design of the current patient-specific knee prosthesis does not follow the tibial plateau curvature.

## 1. Introduction

Knee arthroplasty is becoming more common as the prevalence of osteoarthritis and life expectancy increases. [1]. Depending on the extent of joint disease, total knee arthroplasty (TKA) or unicompartmental knee arthroplasty (UKA) can be applied and both approaches have advantages in long-term survivorship and functional outcomes [2,3]. Orthopedic surgeons have difficulty deciding a method of treatment for young and active patients [4]. UKA has many theoretical advantages, such as the preservation of bone stock, a more rapid recovery and rehabilitation, and better functional results. Furthermore, UKA has the advantages of the conservation of anterior and posterior cruciate ligaments and normal kinematics [5]. However, UKA is required to a delicate surgical method, and in some cases, malalignment of the components has caused poor post-operative functions and early revisions [6,7]. To improve the accuracy of implanted component alignment, computer-assisted surgery systems have been developed [8,9]. In addition, TKA becomes a gold-standard treatment for patients with knee joint arthritis. It can provide them with the relief of pain, restorative function, and overall satisfaction [10]. Reports have demonstrated good medium- to long-term survivorship after TKA. However, in the elderly and young population, active TKA patients are on the increase and efforts have increased to study long-term survivorship and better clinical outcomes [11]. The common reasons for patient dissatisfaction with regard to TKA include preoperative functional loss, limited range of motion, abnormal sensations, crepitation, and residual pain in the knee [12]. Among the reasons for dissatisfaction with outcomes reported by numerous patients, the three most common are pain, stiffness, and limited function [13]. Demographic factors, including age body mass index (BMI) and gender, are believed to also influence patient outcomes [14]. The geometries of conventional TKA are designed by anthropometric population standards, which accommodate common knees and a wide range of subtle anatomical variations. Although in most cases, it can achieve an adequate fit, mismatch occurs with a certain frequency, and this can affect the clinical outcome. Mahoney et al. assessed 437 TKA cases and found that femoral overhang of 3 mm occurred in 57% of the cases. [15]. An oversized anteroposterior (AP) of the femoral components shifts the flexion gap, causing anterior overstuffing or tightness that increases the risk of post-operative patellofemoral symptoms [10,16,17]. The femoral or tibial overhang on the medial or lateral can increase the risk of soft-tissue impingement. [9,10,18]. Furthermore, studies have shown that the major cause of pain is errors of internal rotation of TKA components, especially those of the tibia, and lead to functional deficits after TKA. [19,20]. Maximized fit and coverage of the exposed tibia often cause malrotation. Accordingly, surgeons have a tendency to downsize tibial components to achieve the correct rotation of the component without overhang [21]. However, these results bring about tibial undercoverage of the cut surface, and it has been hypothesized to provide to subsidence, increased osteolysis from wear debris, and component loosening [22,23]. These findings in the literature review and risk factors to affect total knee arthroplasty are summarized in Table 1.

Therefore, surgeons are often constrained to choose between achieving an optimal fit and obtaining proper tibial rotation. This compromise has been reported to be correlated with clinically significant pain of the knee two years after the operation and to be related to the use of larger TKA prostheses, short patient stature, and the female gender. In addition, it is impossible that conventional TKA geometries accommodate a variety of ethnic anthropometric variations. Mediolateral to anteroposterior (ML/AP) ratio of Asian patients differs from that of Caucasians. In the Korean population, it was shown that smaller knees exhibit a larger ML/AP ratio, and a smaller ML/AP ratio is exhibited in larger knees [19].

This may lead to ML undercoverage and overcoverage, respectively, when TKA components based on the femurs of Caucasian patients are used. Moreover, the prostheses of conventional TKA cannot cover the wide spectrum of ethnic tibiofemoral morphotypes [25]. Mismatch in the angles of the sulcus between the native knees and prostheses has been proved, however, it is not clear to what extent this is true for other patellofemoral ratios [25,26]. In recent studies, it was shown that four of the five measured trochlear morphometric parameters relied significantly on ethnicity, whereas only two of the measured parameters relied on gender [26]. Such research, including anatomical analysis, provides standards for the suitable design of femur and tibia components, taking into account gender or patient-specific differences in the Asian population [24,27,28,29,30,31].

In recent years, many technologies have been introduced to provide better functional outcomes after knee arthroplasty. The creation of customized implants for an individual patient has become possible with advanced technology. Patient-specific UKA or patient-specific TKA has the advantage of reducing the rate of underhangs/overhangs to perfect coverage of the resected bone, theoretically [32]. The attention of patient-specific implants (PSIs) has increased for the purpose of increasing implant durability and decreasing or maintaining the associated cost. PSIs develop an alignment guide to fit each component of a patient’s unique anatomy using magnetic resonance imaging (MRI) or computed tomography (CT) [33,34,35,36,37]. The crucial questions are related to how outcomes can be improved and what can be made better with customization. Further investigations into the biomechanics, patient recovery process, cost, and true efficacy of PSI surgical options are required.

## 2. Motivation and Literature Search on Patient-Specific Knee Arthroplasty

For this literature review, the PubMed, MEDLINE, Embase, and Scopus databases were searched for related studies published prior to 1 February 2020. The following keywords were used: “customized knee prosthesis” and “patient-specific knee prosthesis”. Patient-specific instrument techniques were excluded. Two researchers independently reviewed all of the potentially eligible abstracts and full texts. If any disagreement occurred, a consensus was reached through consultation with a third researcher. The search focused on studies published in English.

## 3. Review of Patient-Specific Knee Arthroplasty

### 3.1. The Fits of Patient-Specific Knee Arthroplasty

One of the primary purposes of the patient-specific knee arthroplasty design is to reduce malalignment and to make an implant fit optimally without a size mismatch in order to minimize associated complications or implant failure (Figure 1).

A recent study showed that 23% (544 of 2367) of AP measurements and 25% (592 of 2367) of ML measurements exhibited a 6 ± 3 mm mismatch between the patient’s bony anatomy and the dimensions of the femoral component with a modern standard symmetric TKA or an asymmetric TKA design [38]. However, Kurtz et al. showed that bone resection of the femur and tibia was significantly reduced when a customized knee arthroplasty with a patient-specific instrument was used. When assessed volumetrically with a computer-aided design image, conventional implants involved the resection of 12–49% more bone than PSIs, depending on the size of the implant used. In addition, Carpenter showed the advantage of PSIs, which provide superior cortical bone coverage and fit by minimizing the overhang and undercoverage observed with conventional implants [39]. In addition, such a patient-specific knee arthroplasty is advantageous in lateral UKA. Traditionally, the tibial plateau is rounder in the lateral side than in the medial side. The tibial implants are in discord with the AP/ML ratio of the tibial plateau, so surgeons need to use maneuvers to make amends for these shortcomings. The tibial component is moved relatively medially, not covering the most lateral aspect of the tibial plateau, and the femoral component is intentionally moved as laterally as possible [34]. The lateral condyle is smaller, and oversizing of the femoral component can lead to patellofemoral impingement [34]. Demange et al. showed that patient-specific lateral UKA performed better in tibial coverage, radiological, and short-term clinical results compared to conventional lateral UKA [40]. In a patient-specific knee arthroplasty, a commercial program uses an MRI or CT scan to perform patient-specific measurements of the complete lower extremity to optimally guide the operative plan and the specific bone cuts. Ideal joint alignment theoretically results from the patient’s postoperative mechanical leg axis, coronal alignment of the femoral and tibial components, and sagittal alignment of the femoral and tibial components [41]. Ivie et al. suggested that reliable reproduction of the limb mechanical axis may accrue from patient-specific guides with patient-specific TKA when compared with intramedullary instrumentation in standard TKA [42]. Other studies indicated that customized TKA with PSI showed results comparable to those of computer-assisted surgery systems. This technology enables improved implant fit and restoration of the patient’s J-curves by offering accurate neutral coronal mechanical alignment; however, further investigation is required [11,43]. Schroeder et al. also found that customized TKA has the benefit of achieving optimal tibial rotation while maintaining a proper fit compared with conventional TKA [43]. This trend can also be found in patient-specific UKA [44]. Koeck et al. observed the advantage of patient-specific fixed-bearing UKA of avoiding the malpositioning of components and restoring the axis of the leg, thereby ensuring maximal tibial coverage [44].

### 3.2. Clinical Outcome of Patient-Specific Knee Arthroplasty

Customized knee arthroplasty improves kinematic function and consequently, patient satisfaction [45]. Zeller et al. reported that patient-specific TKA has a kinematic similar to a normal knee; therefore this technology of customized implant can provide more benefits than conventional TKA [46]. Another previous study showed that PSIs were expected to show a statistically significant decrease in blood loss and length of hospital stay, even though these variables did not have significant clinical differences between the two groups studied [47]. A recent study also showed that Knee Society scores were significantly higher in the patient-specific TKA group and that the functional scores translated to better basic daily functions [48]. Additionally, the patient-specific TKA group achieved a higher global patient satisfaction rate [48].

Wang et al. showed that patients who underwent customized bi-compartmental knee arthroplasty exhibited better strength and mechanics in daily activities [49]. A recent study also showed that customized bi-compartmental knee arthroplasty allowed an accurate fit of the implants and provided significant improvements postoperatively with a higher satisfaction during the short- to medium-term follow-up [50]. This novel customized bi-compartmental knee arthroplasty is resurfacing and does not require faceted cuts of 10 mm with a thickness of 3 mm, thereby preserving bone stock for a future revision [50]. This may be an alternative method for young and active patients with bi-compartmental osteoarthritis (OA), although research into a longer-term follow-up is necessary. Recently, a multicenter and prospective study reported patient outcomes and safety profiles (revision rates) with a monolithic customized bi-compartmental knee arthroplasty [51]. The study showed that customized bi-compartmental knee arthroplasty compared favorably to both published scores and revision rates for formerly available monolithic conventional bi-compartmental knee arthroplasty. Customized bi-compartmental knee arthroplasty offers the solution of a feasible and patient-specific monolithic implant to surgeons for patients with bi-compartmental disease that can be treated by unicondylar and patellofemoral joint or bi-cruciate-retaining TKA surgeries [51].

Another advantage of patient-specific knee arthroplasty is the efficiency of surgical procedures, saving time and money, and reducing medical complications [52,53]. O’Connor and Blau proved that customized implants of TKA can accomplish considerable savings compared with conventional Medicare program implants and primarily has lower initial average procedure costs and less post-operate for inpatient services and skilled nursing facilities [54]. However, patient-specific knee arthroplasty does not always provide positive clinical outcomes. Talmo et al. observed a relatively high percentage of aseptic loosening, femoral component looseness, in the short-term and medium-term follow-up period with patient-specific UKA [55]. However, they also stated that further studies with larger numbers of customized UKA from multiple institutions may help to verify these findings. In addition, customized TKA can improve kinematics for TKA patients [56]. White and Ranawat showed that patient-specific TKAs were associated with higher manipulation rates compared with conventional TKA [57]. Recently, Kumar et al. suggested that careful attention to the surgical technique is critical in the optimization of implant survivorship with the customized TKA design [58]. However, a recent study showed that customized TKA resulted in a manipulation-under-anesthesia rate that is consistent with that reported in the literature for all designs [59,60]. A study by Sanz-Rui et al. showed that patient-specific instrumentation may improve component alignment in the learning curve of surgeons, thereby achieving functional results similar to those of more experienced surgeons using a conventional procedure [61]. In addition, the patients showed significant improvements in the range of motion and Knee Society scores [59]. Meheux et al. retrospectively compared patient-reported outcome scores, radiographic outcomes, and complication/revision rates between patient-specific TKA and conventional TKAs [62]. In this study, the manufacturer of the patient-specific prosthesis modified the locking mechanism and the design of the tibial insert because early failures were reported. The study was classified into patient-specific design (PSD)-1, which was a group of patients with early patient-specific design implants and PSD-2, which was the other group with modified implants [62]. Ultimate failure was observed in the PSD-1 group with failures in tibial subsidence and the polyethylene locking mechanism. PSD-2 showed better early Knee Society function scores, shorter hospital stays, smaller declines in hemoglobin, radiographic alignment, and no failures compared with PSD-1 and conventional TKA.

### 3.3. Biomechanical Effects of Patient-Specific Knee Arthroplasty

Various researchers have performed in vitro experiments and computational simulations to evaluate the biomechanical effects of patient-specific knee arthroplasty. By using a computational simulation for design, Harrysson et al. showed that their proposed custom femoral component has advantages over the conventional femoral component [63]. As the articulating surface is similar to the shape of the distal femur, there is no need for gait change or the resurfacing of the patella. Due to the resulting stress distribution, bone remodeling is even and the risk of premature loosening may be reduced [63]. Because the bone-implant interface can accommodate anatomical abnormalities at the distal femur, it reduces the need for surgical interventions and fitting of filler components [63]. As the bone-implant interface is customized, approximately 40% less bone needs to be removed [63]. Through cadaveric experiments, Patil et al. showed that patient-specific TKA generates kinematics that more closely resemble the kinematics of the normal knee, compared with the conventional implant [64]. The more normal kinematics achieved with patient-specific TKA may result in the resolution of many of the clinical problems observed with conventional TKA, such as anterior knee pain, mid-flexion instability, reduction in the range of flexion, and incomplete return of function [64]. Using computational simulations, van den Heever et al. showed that patient-specific UKA results in lower contact stresses at the tibiofemoral joint and also in lower displacement rates compared with the conventional UKA design [65]. In addition, Kang et al. reported that patient-specific UKA provided mechanics closer to those of the normal knee joint and that the decreased contact stress on the opposite compartment may reduce the overall risk of progressive OA compared with that of conventional UKA [32]. Koh et al. showed that restoration of the normal geometry of the knee joint in patient-specific bi-cruciate-retaining TKA and preservation of the anterior cruciate ligament can lead to an improvement in kinematics compared with conventional posterior C and bi-cruciate-retaining TKA [66]. Recently, Wang et al. showed that a patient-specific TKA design with both cruciate ligaments retained could move more naturally; however, improvement in the patient-specific TKA is still required to reduce the large tibiofemoral compressive force after 50 knee flexions [67].

## 4. Discussion and Future Direction

As TKA became more popular with the loss of joint function and knee OA treatment, investigating the method for accuracy, reproducibility, and the effectiveness of the procedure has become crucial in modern orthopedic studies with patient-specific TKA showing potential [68,69]. As previously mentioned, developing a custom knee arthroplasty requires advanced imaging techniques, either CT or MRI, to obtain the patient’s specific anatomy of the hip, distal femur, proximal tibia, and ankle [70]. Through this process, individualized implants can be developed by matching the geometry of the patient’s tibial plateau and femoral condyles. This model was designed to maintain knee stability at all movement levels while simultaneously maintaining a neutral mechanical axis and maintaining a constant relationship between joint points across the range of motion [71,72]. The implant that considers the coronal radii of the trochlear groove and condyles is designed to reduce polyethylene wear. This design can lead to the lowest possible contact stresses [73]. In addition, Buller et al. found that custom knee joint surgery needs thinner cross-sections to perform rotation and a custom-fit, allowing for bone stock preservation. This process permits the replacement (from a surface perspective) of exactly what was cut (from a shape perspective). On the tibial side, different sizes of medial and lateral inserts have the advantage of matching the femoral offset while restoring the tibial plateau angle of the normal knee [70]. This study also shows that for the facilitating of the natural femoral rollback during the knee flexion, the medial insert is more conforming [70]. To achieve this level of precision, implant systems of custom TKA are manufactured with a custom-cutting instrument for placing the components exactly. In the prior TKA approach, surgeons had limitations related to kinematics due to anatomy variations of the patient. The mechanical axis is restored by custom TKA through a method requiring perpendicular cuts. By using an anatomical implant with an asymmetric thickness, restoration of the joint line and the normal knee kinematics is possible. Fundamentally, custom TKA merges the benefits of two major strategies of condylar knee design.

Currently, patient-specific knee arthroplasty systems are manufactured by ConforMIS in the United States, BodyCAD in Canada, and Symbios in Switzerland. In patient-specific TKA, the design of the tibial insert is developed using the articular geometry obtained from the femoral component [33]. The complete design of patient-specific UKA is based on the variability in the femoral component’s coronal curvature, which may cause point loading in specific flexion angles when using a curved tibial insert [74]. To resolve the problem, a flat polyethylene tibial component has to be considered in conjunction with a constant coronal curvature of the femoral component to ensure constant loading conditions over a large surface regardless of the flexion angle when the tibial insert is designed [74]. In other words, the tibial insert in the current patient-specific knee arthroplasty design cannot perfectly preserve the tibial plateau curvature. In the native knee, however, the medial tibial plateau has slightly dished geometry and the lateral is convex (Figure 2) [75].

Additionally, the medial and lateral menisci are considerably different in their biomechanics [76,77]. The medial meniscus has significantly less movement than the lateral meniscus because of its attachment to the medial collateral ligament and larger insertion areas. Thus, the medial meniscus affects joint stability more than the lateral meniscus, which closely follows the AP excursion of the femur [76,77]. The dished medial plateau and the greater stability of the medial meniscus restrict the AP motion and posterior rollback of the medial femoral condyle. On the contrary, the convex lateral plateau and the lateral meniscus mobility enable a greater range of AP motion with a greater posterior rollback of the lateral femoral condyle. Therefore, in high flexion activities, such as a deep-knee-bend, the knee shows an overall medial pivot motion with a greater rollback of the lateral femoral condyle [78,79]. However, it is important to note that the medial tibia does not completely constrain the medial femoral condyle. Accordingly, during limited flexion activities, such as climbing stairs, the AP motion of the medial condyle may be similar to the motion of the lateral condyle, although the knee shows an overall medial pivot over its full range of motion [80,81]. For the solution of the kinematic limitations of contemporary implants, a novel design process was introduced to develop anatomy-mimetic articular surfaces directly from in vivo knee joint motion [82,83,84].

Varadarajan et al. showed that an anatomy-mimetic TKA (Figure 3) more closely mimicked the normal kinematic patterns than a conventional TKA [82]. In particular, the anatomy-mimetic TKA showed medial pivot motion, whereas conventional TKAs showed abnormal motion including lateral pivot or no pivot and paradoxical anterior sliding during deep-knee-bend and chair-sit motions [83]. In addition, other groups aimed to define the patient-specific design and to evaluate the kinematic results [85,86,87]. The studies showed that a customized TKA provided a motion pattern closer to the normal target than expected, not only during a gait cycle but also as the knee flexes to higher degrees during squatting. The major design features of customized TKA take into account the location and orientation of the flexion and pivoting axes, the trace of the contact points on the tibia, and the radii of the guiding arcs on the lateral condyle [85]. Recently, Koh et al. performed computational simulations to investigate whether the natural knee kinematics were preserved with respect to tibiofemoral conformity and the effect on wear in patient-specific knee arthroplasty [88,89,90,91,92,93,94,95,96]. Koh et al. showed that anatomy-mimetic cruciate-retaining (CR) patient-specific TKA provided the closest-to-normal kinematics; however, even anatomy-mimetic CR patient-specific TKA could not restore the normal knee biomechanics owing to the absence of the anterior cruciate ligament [89]. This trend was also observed in posterior-stabilized (PS) patient-specific TKA [88]. Anatomy-mimetic PS patient-specific TKA showed the closest-to-normal kinematics in the deep-knee-bend condition [88]. Studies also indicated that the convex post design and subject anatomy-mimetic tibiofemoral surfaces provided the closest-to-normal knee kinematics [92,93]. The studies suggested that the post-cam design and tibiofemoral surface conformity should be considered in conventional and customized TKA [92,93]. Recently, Koh et al. investigated the extent to which normal knee kinematics were preserved with respect to the tibial insert design in mobile-bearing medial patient-specific UKA [90]. Whereas all current existing patient-specific UKA designs follow a fixed-bearing system, the authors showed that by replacing the anatomy-mimetic design with a mobile-bearing design, the natural knee kinematics were preserved during gait and deep-knee-bend motions [90]. These results show the importance of tibiofemoral conformity in preserving the native knee kinematics in patient-specific mobile-bearing UKA [90]. This importance was proved in not only mobile-bearing but also fixed-bearing medial patient-specific UKA [97,98].

The etiology of failure mechanisms is important in the delivery of appropriate care [56]. Compared with a hip prosthesis, wear and osteolysis in a knee prosthesis have less effect on long-term survival [99]. Wear continues to be an issue, however, owing to the incongruent and instability of the knee joint that often requires revision surgeries. Wear is also one of the important factors in patient-specific TKA. Koh et al. not only studied the natural knee kinematics in patient-specific knee arthroplasty but also expanded their study to wear performance [91,95]. Their study showed that tibiofemoral articular surface conformity should be considered in customized PS TKA designs [95]. Different wear performances were observed with respect to different tibiofemoral conformities. Although customized anatomy-mimetic PS TKA showed an inferior wear performance compared to customized medial pivot conformity PS TKA, it was better in terms of kinematics; thus, its functionality might be improved by optimizing the tibiofemoral articular surface conformity [95]. In addition, such a TKA should be carefully designed because any changes may affect the post-cam mechanism [95].

This trend was also found in CR patient-specific TKA [100]. A previous study showed that conformity changes in the femoral and tibial inserts influence the wear performance in CR patient-specific TKA [100]. Kinematics and contact parameters should be considered to improve wear performance in CR patient-specific TKA [100]. The conformity modification in the tibiofemoral joint changes the kinematics, contact parameters, and wear performance [100]. However, anatomy-mimetic CR patient-specific TKA did not show the best wear performance. A recent study showed the potential to reduce the wear in patient-specific TKA using design optimization and parametric three-dimensional finite-element modeling [101]. This study demonstrated that the design optimization of patient-specific TKA can improve the wear performance with conserved kinematics, providing a potential method of increasing the lifespan of patient-specific TKA [101]. Another recent study showed that fixed-bearing patient-specific UKA showed increased conformity and provided improvements in wear but resulted in limited kinematics [91]. Therefore, increased conformity should be avoided in fixed-bearing patient-specific UKA designs. In the article, the use of flat or anatomy-mimetic tibial insert design in patient-specific UKA was suggested [91]. Another study also showed that increased conformity produced more wear even in mobile-bearing patient-specific UKA; however, highly cross-linked mobile-bearing polyethylene inserts can also improve wear performance [102]. These results provide improvements in design and materials to reduce wear in mobile-bearing UKA [102].

As previously mentioned, the anatomies of the tibial plateau of the medial and lateral sides are different. Patient-specific UKA is advantageous for the lateral side because the design of common UKA is considered for the larger volume of patients [103,104]. In a recent study, convex tibial insert design that mimicked the anatomy showed kinematics similar to those of the native knee in lateral patient-specific UKA.

In this review, improved kinematics were observed with patient-specific knee arthroplasty compared with conventional knee arthroplasty. One reason for this finding could be the ability of the patient-specific knee arthroplasty implant to regenerate the patient’s sagittal J-curves with better accuracy compared with the conventional knee arthroplasty, leading to a higher likelihood of a stable knee in flexion. The theoretical advantages of a complete patient-specific implant and instrumentation extend beyond the aforementioned description. It can potentially reduce the operative and set-up times, as well as the use of operating rooms and hospital spaces [96,105,106]. The procedural time can be reduced with a complete patient-specific system that includes all instrumentation and a patient-specific implant by eliminating several time-consuming steps. When the instrumentation and implants are completely specific to the patient, the implant sizing, rotation, and positioning are predetermined. These implant attributes can be either based on design rules and standards or could be customized according to the surgeon’s preferences. However, this new technique requires an additional CT or MRI scan from the hip to the ankle, which results in additional radiation exposure. Another disadvantage of this implant is the long manufacturing time (approximately six weeks) [34]. Nevertheless, a similar delay has existed in manufacturing dental crowns and other implants, as well as in transplant surgery. In most joint surgeries, the operation is not scheduled immediately but rather within a period of six to eight weeks after the initial diagnosis, considering the time needed for the production and delivery of the implants and instruments [34].

However, as previously mentioned, current patient-specific knee arthroplasty does not preserve the tibial plateau anatomy perfectly. In addition, there is currently no patient-specific TKA that preserves the anterior cruciate ligament. A previous study showed through dynamic simulations that an anterior cruciate ligament-substituting design that retains the native posterior cruciate ligament showed important kinematic improvements over a CR TKA design [107]. In particular, the abnormal posterior femoral shift and paradoxical anterior sliding in low knee flexion observed with the CR implants were addressed with the anterior cruciate ligament-substituting design through the replacement of the native anterior cruciate ligament by a substituting post [107]. Recently, a conventional prosthesis designed with a concave medial tibial insert and a convex lateral tibial insert was launched [108], and long-term clinical results in this regard are yet to be obtained. A study on a preserved tibial plateau anatomy in patient-specific UKA and TKA is necessary, as several previous studies have demonstrated that the preservation of tibial plateau anatomy leads to better kinematics in patient-specific knee arthroplasty.

## 5. Conclusions

In conclusion, patient-specific knee arthroplasty is more advanced than a conventional anatomical approach to knee arthroplasty. By using state-of-the-art technologies, an anatomical patient-specific knee arthroplasty implant is generated based on modern imaging technologies to provide an individual fit, optimize coverage, preserve the individual J-curves of all three knee compartments, and restore the distal femoral individual offset, thereby decreasing the ligament balancing requirement. In addition, with its increasing popularity in patient-specific knee arthroplasty, customized technology has potential applications in several orthopedic surgical procedures. Although the efficacy of patient-specific knee arthroplasty is still controversial, the theoretical alignment and accuracy of PSIs provide potential advantages compared with the conventional implant. However, there are some considerations for applying patient-specific knee arthroplasty. For meaningful patient-specific knee arthroplasty, an accurate analysis of the patient’s knee image should be performed. Long-term studies are necessary to determine whether the early biomechanical and clinical advantages are crucial in patient-specific knee arthroplasty. In addition, the currently used tibial insert design for patient-specific knee prostheses does not follow the tibial plateau curvature. In the future, the tibial plateau curvature of each individual patient should be considered when designing and manufacturing patient-specific knee prostheses.

## Figures and Tables

**Figure 1 jcm-09-01559-f001:**
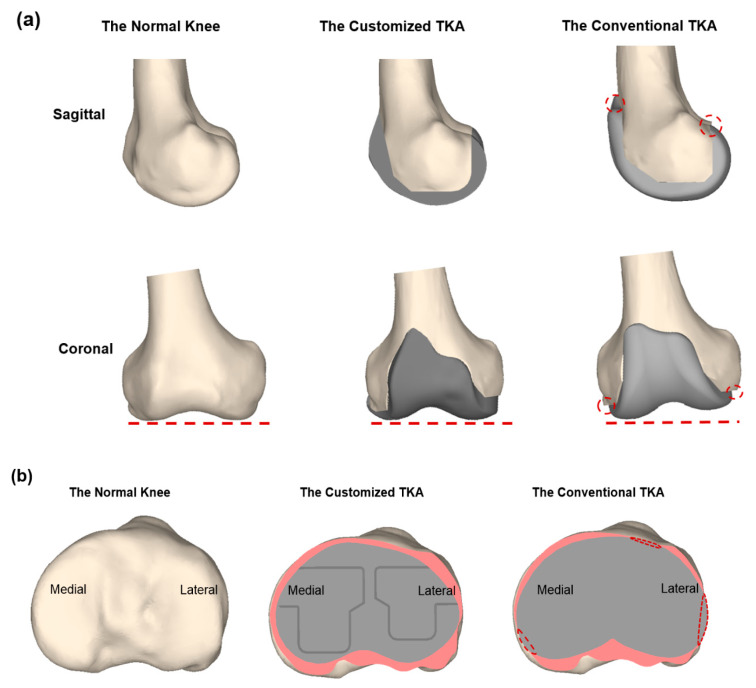
Comparison of a normal knee, conventional TKA, and customized TKA for (**a**) femur; (**b**) tibia.

**Figure 2 jcm-09-01559-f002:**
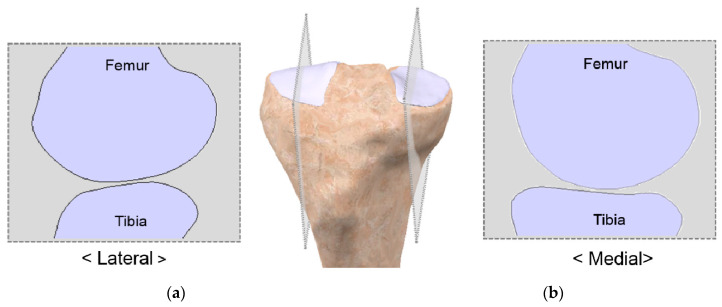
Native knee model showing (**a**) the convex lateral plateau; (**b**) dished medial plateau asymmetry.

**Figure 3 jcm-09-01559-f003:**
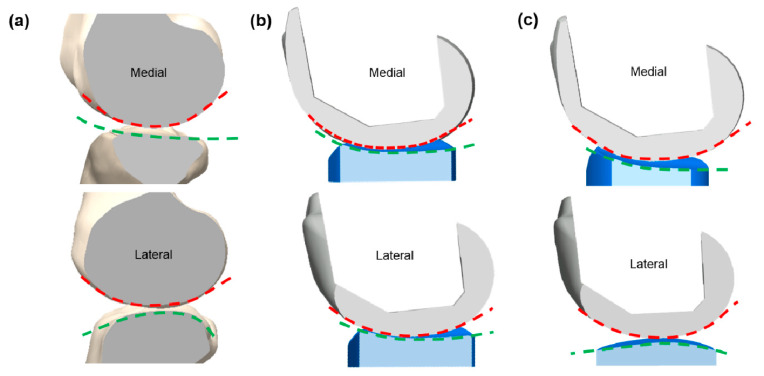
Comparison of tibiofemoral conformity of (**a**) normal knee; (**b**) femoral component derived from customized TKA and tibial insert derived from femoral component; (**c**) fully anatomy-mimetic customized TKA.

**Table 1 jcm-09-01559-t001:** Summary of findings of literature review and risk factors affecting total knee arthroplasty.

Authors	Year	Risk Factor	Study Design	Findings
Motesharei et al. [9]	2018	Implant alignment	Comparison of a traditional and robotic UKA system	Improve method of the accuracy of implant alignment
John et al. [10]	2012	Implant alignment	Comparison of a conventional and patient-matched instrument system	The accuracy of mechanical alignment for the patient-matched instrument system
Mahoney et al. [15]	2010	Overhang	Gender comparison	Occurring more often and with greater severity in women
Shrinand et al. [16]	2000	Overhang	Anthropometric population	Design of the prosthetic components for the Indian population
Ranawat [17]	1986	Implant dislocation	Clinical follow-up	The effect of the patellofemoral joint in TKA
C.W Ha et al. [19]	2012	Implant size mismatch	Anthropometric population	Design to improve the fit of TKAs for the Asians.
D. Nicoll et al. [20]	2010	Implant alignment	Clinical follow-up	The effect of the rotational alignment
R.A. Burger et al. [21]	1998	Implant alignment	Case study	Rotational malalignment of TKA causes loosening, pain, and infection
S. Martin et al. [22]	2013	Implant placement	Case study	The effects of implant placement on TKA
Y.G. Koh et al. [24]	2017	Implant size mismatch	Anthropometric population	Difference between gender in the medial-lateral condyles

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
