# Peer review of "Biomechanical and Clinical Effect of Patient-Specific or Customized Knee Implants: A Review"

_jcm, 2020, doi:10.3390/jcm9051559_

Round 1

Reviewer 1 Report

The manuscript of second version is good condition.

Author Response

I have corrected for similar sentence in the paper.

Reviewer 2 Report

Accepted

Author Response

I have corrected for similar sentence in the paper.

This manuscript is a resubmission of an earlier submission. The following is a list of the peer review reports and author responses from that submission.

Round 1

Reviewer 1 Report

An interesting paper which details the relevant literature for customized knees.  The intro lacks a clear purpose.  A few concerns of the methods used.  I would separate UKA, bicondylar UKA, partial knee etc from TKA.  The literature is varied on these topics and confuses the reader.  A table listing the relevant studies with the appropriate headings such as outcomes (objective and subjective), followup, complications, and level of evidence would be helpful.  Line 110 has a typo in the results being presented.  The nomenclature used in this paper is confusing.  The use of the term KA is universally known as "kinematic alignment" not Knee arthroplasty.  PS is known as posterior stabilized, it is not usually used for patient specific instrumentation or PSI.  The title is confusing by using the term Patient-Specific, of which the authors are referring to a type of customized TKA, and not Patient specific instrumentation.  The results section should be broken down into different sections such as fit, function, outcomes etc.  I think this paper is closest to a short primer and should be organized as such.  I see the author has quoted himself 21 times in the references. Listing only those studies with the highest level of evidence would make the paper more impactful. 

Reviewer 2 Report

The authors conducted a review to evaluate the clinical and biomechanical efficacy of patient-specific knee prosthesis, explore its future direction, and summarize any published comparative studies. This manuscript does not provide a lot of original and new find among many previous studies. However, this manuscript is considered valuable as a review study, and has value in that it addresses an important theme. Thank you for the opportunity to review this manuscript.

A review / research paper includes a review article, which is itself a single article, and a literature review of prior research as part of the article. This paper advocates the so-called review article. Therefore, it will be clear what has already been clarified and what has not been clarified, and it is expected that the next problem to be solved will become clear.

From the perspective of establishing EBM, it is hoped that the review article will explain the methods and results in a coherent manner. Based on that, expert opinion is also important. Therefore, the concept of QUOROM statement and PRISMA statement is presumed to be helpful in this research, even if it is not a systematic review or meta-analysis. 

1. Authors should provide a more detailed explanation of the review process and criteria. In particular, adding search strategies and flowchart diagrams will greatly help the reader's understanding. Authors should also specify review methods for reproduction and development by other researchers.

2. Authors should provide a more detailed explanation of the results of the review of the article. In particular, scientific knowledge should be summarized for each issue (evaluation) or field (procedure) according to the purpose of research. As with other review papers, it is desirable to add a table summarizing the distribution of safety and efficacy. 

3. The authors conclude this paper by suggesting issues of procedural related to Tibial Component. On the other hand, in the results, although Knee Society function score and tibial subsidence failure were mentioned, the extent and level of them were not clarified. In addition to KSS, KF, and BMI, it is desirable to add incidental information such as HKA alignment.

4. In this paper, there may be carelessness of the first abbreviation (eg PS) or numerical notation (eg a 63 mm mismatch). I encourage the author to recheck, just in case.

Reviewer 3 Report

Comments on Manuscript Number: jcm-756204

Biomechanical and Clinical Effect of Patient-Specific 3 or Customized Knee Implants: A Review

The authors present their review on the use of Patient-Specific and Customized knee implants.

The topic is interesting, dealing with a hot debated field. Brand-new and modern technologies surely represent the future directions of orthopaedic surgery but some aspects have to be strongly considered: the mean follow-up of all studies are not adequate to standard techniques; challenging or very difficult cases ; finally, all orthopaedic surgeons using one or both the mentioned options have reported in their experiences problems or other intraoperative issues that have induced a variation of strategy during surgery, even the aborting of the PS-manner and the conversion to standard knee arthroplasty. Not all such issues have been reported in literature, and consequently what is published seems to be mostly revolutionary.

Such aspects should be clearly highlighted in the paper before considering it viable for publication in JCM.

General overview:

  • The paper is substantially well organized and English style is proper, also by a scientific point of view
  • Some improvements are useful as later mentioned, in order to make the manuscript more readable

Specific remarks:

  • Introduction:

- Lines 35-44: as it is presented, the meaning of such introduction seems a comparison between UKA and TKA, that have definitely different indications. Please, try to better explain such concepts

  • Mat & Met

- Given the several types of techniques studied in this review (PS CR and PS TKA, PS UKA, Customized UKA and TKA), I suggest the use of subchapters with specific headings: it would be better understandable by the readers. Such partition should be also reported in the Discussion section

  • Results:

- As proposed above, structured M6M may be also better evaluated by Tables with similar structured types of implant and technique

  • Discussion:

- See M&M and Results comments

- It is of paramount importance to analyze the results of such new technologies, but also to highlight that not always PS KA is feasible (most on “standard” cases, not in patients with severe malalignment, challenging conditions – as post-traumatic, rheumatic, or haemorrhagic subjects), needing the opening of several decks and the increment of cots; moreover, a customized component may not always be perfectly congruent to the real status of the joint (mostly related to inaccurate or not fully accurate pre-operative imaging study), or the final implant well balanced.

In other words, it should not pass the message that PS and customized knee surgery is the pathway to drive always trough.

I think that the paper has to be considered for publication in JCM after minor revisions.